# Imaging Flow Cytometric Analysis of Stilbene-Dependent Apoptosis in Drug Resistant Human Leukemic Cell Lines

**DOI:** 10.3390/molecules24101896

**Published:** 2019-05-17

**Authors:** Marcin Czop, Anna Bogucka-Kocka, Tomasz Kubrak, Karolina Knap-Czop, Anna Makuch-Kocka, Dariusz Galkowski, Joanna Wawer, Tomasz Kocki, Janusz Kocki

**Affiliations:** 1Clinical Genetics Department, Medical University of Lublin, 11 Radziwillowska Str., 20-080 Lublin, Poland; karolinaknap@gmail.com (K.K.-C.); jwaawer@gmail.com (J.W.); januszkocki@umlub.pl (J.K.); 2Chair and Department of Biology and Genetics, Medical University of Lublin, 4a Chodzki Str., 20-093 Lublin, Poland; anna.kocka@tlen.pl; 3Department of Biochemistry and General Chemistry, University of Rzeszów, 2a Kopisto Al., 35-959 Rzeszów, Poland; kubrak.tomasz@gmail.com; 4Department of Pharmacology, Medical University of Lublin, 4a Chodzki Str., 20-093 Lublin, Poland; anna.makuch@poczta.fm; 5Department of Pathology and Laboratory Medicine, Rutgers Robert Wood Johnson Medical School, Medical Education Building – 212, One Robert Wood Johnson Place, New Brunswick, NJ 08903-0019, USA; galkowd@fastmail.fm; 6Chair and Department of Experimental and Clinical Pharmacology, Medical University of Lublin, 8b Jaczewskiego Str, 20-093 Lublin, Poland; tomasz.kocki@umlub.pl

**Keywords:** imaging flow cytometry, apoptosis, leukemic cell lines, stilbene derivatives, multidrug resistance

## Abstract

**Background:** The natural compounds have been researched extensively as an alternative to the conventional chemotherapy and radiation. Stilbene derivatives appear as a group of therapeutics which deserves special attention. The present study was designed to analyze the effects of stilbene derivatives on drug resistant human leukemic cells. The aim of this work was to evaluate the apoptotic effect of stilbene derivatives in various concentrations on leukemic cells (LC) with and without resistant phenotype. **Methods:** Human acute promyelocytic leukemia (APL) cell lines (HL60, HL60/MX1, HL60/MX2) and acute lymphoblastic leukemia (ALL) cell lines (CEM/C1, CCRF-CEM) were studied. T-resveratrol, piceatannol, rhaponticin, deoxyrhaponticin, pterostilbene were used to stimulate apoptosis. Mitoxantrone (MIT) was applied to induce drug resistance. **Results:**
*t*-Resveratrol (RES), deoxyrhaponticin (D-RHAP), rhaponticin (RHAP), pterostilbene (PTER), and piceatannol (PIC) influenced viability and induced apoptosis in all investigated cell lines. **Conclusions:** Our results confirmed that RES, PIC, RHAP, D-RHAP, and PTER are essential therapeutic compounds with anticancer activity exhibited by induction of apoptosis in leukemic cells with and without resistant phenotype. Stilbene-induced apoptosis in HL60/MX1, HL60/MX2, CEM/C1, and CCRF-CEM leukemia cell lines have been presented in very few studies so far and our research is an important contribution to the investigation of these substances.

## 1. Introduction

Cancer still presents as one of the most substantial healthcare problems in many, especially, low- and middle-income countries. The global burden of malignancy has significant economic and social impact on societies. Treatment of cancer is still in large primarily based on radio- and chemotherapy. Novel methods, like checkpoint inhibitors and cancer immunotherapy, especially in the light of FDA clearance of the first CAR T-cell therapy (tisagenlecleucel—Kymriah) for certain population of children and adults with advanced leukemia, paved the road to further research in this direction. Although this new methods may appear as game-changing treatment, the side effects may be multiple and potentially very serious. This type of therapy is also connected with significant and, in many cases prohibitive, treatment expenses leading to so called ‘financial toxicity’. For these reasons, there is a great need for good alternatives to existing therapies.

Another considerable issue associated with cancer treatment is drug resistance. Cancer chemotherapy has become increasingly complex; still, the cure ratio is not satisfactory. Resistance to treatment may result from a variety of factors, individual genetic variations in patients and in tumor cells, even those from the same tissue of origin, and acquired resistance which has also become more prevalent. The most common mechanism for acquisition of resistance is expression of energy-dependent transporters that detect and expel anticancer drugs from cells. This process takes place with the participation of ATP-dependent transporters. ATP binding cassette (ABC) transporters are a family of membrane proteins, characterized by homologous ATP-binding, and large, spanning transmembrane domains [1,2,3,4]. 

Lack of sensitivity to drug-induced apoptosis and induction of drug-neutralizing mechanisms may also play a role in anticancer drug resistance. Xenobiotic-metabolizing enzymes usually mediate detoxification but can also form reactive intermediates causing toxicity. Cytochrome P450s acts in the phase I with CYP1, CYP2, CYP3, and CYP4 families dominant in the process of xenobiotic metabolism. Phase II metabolism is governed by glutathione S-transferases, Uridine 5′-diphospho(UDP)-glucuronosyltransferases, sulfotransferases, N-acetyltransferases, and epoxide hydrolases, producing excretable hydrophilic metabolites. Transporters in the liver and intestine remove lipophilic compounds and metabolites from cells [5]. In addition, reduced apoptosis, DNA damage repair mechanisms, and altered drug metabolism may lead to drug resistance as well [6,7]. 

Acute lymphoblastic leukemia (ALL) is a malignancy of lymphoid progenitor cells. ALL affects both children and adults. Childhood acute lymphoblastic leukemia (ALL) is curable with chemotherapy in approximately 90% of patients. The precise mechanisms of pathogenesis associated with ALL are not known. Only less than 5% of ALL cases are associated with inherited syndromes like, Down syndrome, Bloom’s syndrome, ataxia telangiectasia, Klinefelter syndrome, or Nijmegen syndrome. Other contributors are ionizing radiation, exposures to certain chemicals, and certain viral infections. Acute lymphocytic leukemia is more common in whites than in African Americans, but the reasons for this are not clear [8,9].

Before the introduction of the vitamin A derivative all-trans retinoic acid (ATRA), acute promyelocytic leukemia (APL) was considered as one of the most fatal forms of acute leukemia associated with poor outcomes. With ATRA treatment and subsequently administering it in combination with anthracyclines, and arsenic trioxide (ATO) regimens, complete remission reached 90% and cure rates 80%. APL is characterized by balanced chromosomal reciprocal translocation t(15:17), which results in the fusion between promyelocytic leukemia (PML) gene and retinoic acid receptor α (RARα) gene. Of all types of AMLs, APL has the biggest ratio of curability. Significant incidence of early death in APL is associated with bleeding diathesis [10,11]. 

The search for alternative anti-cancer drugs derived from natural sources is still considered as one of the priorities in cancer therapy, bearing in mind rapid development of resistance to chemotherapeutic drugs and high toxicity with sometimes deleterious side effects. 

Diet rich in fruit and vegetables has been proven to ameliorate and prevent some of the modern diseases. Antioxidants in fruits and vegetables provide a considerable benefit in reducing disease incidence. Cancer specific cytotoxicity have been reported with the use of flavonoids and stilbenes. Stilbenes are compounds of low molecular weight (210–270 g/mol) which occur in plants naturally, thought to be protective against germs, viruses, and negative effects of ultraviolet radiation. Trans-isomers belong to most often isolated forms of stilbenes. Stilbenes exhibit cardioprotective, anti-inflammatory, and chemopreventive properties with cytotoxic effects on tumor cells [12,13]. 

The best-known and first isolated compound of the stilbene group is *t*-resveratrol (3,4′,5′-trihydroxy-trans-stilbene) (Figure 1 (**1**)), found in red wine, grape skins, peanuts, mulberry tree, berries, and other medicinal plants [12,14,15,16,17]. It exerts complex effect on cell cycle, increasing the expression of p53 protein, inducing apoptosis, inhibiting DNA synthesis by stopping cell cycle in S phase, and reducing the risk of mutation [18,19,20].

Piceatannol (trans-3,4,3′,5′-tertahydroksystilben) (Figure 1 (**2**)) is a derivative of t-resveratrol produced in human cells by metabolism of t-resveratrol by cytochrome P450 (isozymes CYP1A1, CYP1A2) [21]. Piceatannol was first isolated from Euphorbia lagascae in 1984. It is synthesized from *t*-resveratrol as a result of an infection with fungi, bacteria, and during exposure to UV radiation [22]. It occurs naturally in peanuts, the skin of grapes, sugarcane, rhubarb, and berries [23,24,25]. Piceatannol exhibits cytotoxic (antiproliferative) and antioxidant activity [26]

Pterostilbene (trans-3,5-dimethoxy-4′-hydroxystilbene) (Figure 1(**3**)) occurs in the skin of grapes and blueberries. Pterostilbene shows antioxidant, anti-inflammatory, antitumor, antifungal and chemo-preventive activity (CYP1A1 isoenzyme inhibition of cytochrome P450) [12,17,27].

Rhaponticin (3,5,3′-trihydroxy-4′-methoxystilbene 3-*O*-beta-d-glucoside) (Figure 1 (**4**)) and deoxyrhaponticin (3,5-Dihydroxy-4′-methoxystilbene-3-*O*-beta-d-glucoside) (Figure 1 (**5**)) were isolated from a plant of genus Rheum L. It shows antitumor, antithrombotic, antioxidant, and vasorelaxant activity [28,29].

The present study was designed to analyze the effects of stilbene derivatives on drug resistant human leukemic cells. The aim of this work was to evaluate the apoptotic effect of stilbene derivatives in various concentrations on leukemic cells (LC) with and without resistant phenotype using FlowSight Imaging Flow Cytometer.

## 2. Results

A conventional trypan blue assay was used to evaluate the effect of stilbene derivatives on the viability of HL60, HL60/MX1, HL60/MX2, CEM/C1, and CCRF-CEM leukemia cell lines. Trypan blue selectively stained dead cells that lost the integrity of the cell membrane. Incubation with all stilbene derivatives reduced cell viability compared with non-exposed cells in a concentration dependent manner (Figure 2). 

Based on the constructed dose–response curves (Figure 2) IC_50_ doses were calculated for each compound against tested cell lines (Table 1). The obtained values varied depending on the cell line and stilbene derivative tested, and ranged from 4.57 µM for CCRF-CEM cell line exposed to piceatannol to 54.09 µM for HL60 cell line exposed to resveratrol.

The most effective cytotoxic activity was observed for PTER and PIC on both ALL and APL cell lines, while RES and D-RHAP showed lowered activity (Table 1). It was noted that cells of the CCRF-CEM line (non-resistant) are more sensitive to the effects of the stilbene derivatives tested compared to CEM/C1 cells (with a resistance phenotype). This effect was not observed in the case of cytotoxic activity of the studied stilbene derivatives on the HL60 line and its resistant derivatives HL60/MX1 and HL60/MX2. 

The analysis of constructed dose–response curves presenting the cell viability in relation to the concentrations of stilbene derivatives revealed that after initial decrease, some stilbene derivatives, at certain concentration, induced increased cell viability.

This phenomenon was observed after HL60 and HL60/MX2 cell lines were treated with piceatannol at 10–20 µM concentration, CEM/C1 cell line exposed to t-resveratrol at 20–40 µM concentration, CEM/C1 cell line exposed to pterostilbene at 20–30 µM concentration and HL60/MX2 cell line treated with deoxyrhaponticin at 30–40 µM concentration.

To evaluate apoptosis, flow cytometry with imaging in real time was used. This method gives us an opportunity to observe cells in real time. The analysis was performed using Annexin-V-FITC, PhiPhiLux-G_1_D_2_ (caspase-3-FITC antibody), and propidium iodide (PI). FlowSight cytometer enables analysis of cell morphology, the instrument saves images of each cell, allowing for detection of specific morphological changes of the investigated process. Altered cell morphology was confirmed by investigating changes in cell biochemistry.

Caspase-3 is mainly an apoptosis marker. This protein is involved in activation cascade of caspases responsible for execution of apoptosis. These proteinases have been described as factors participating in apoptotic DNA fragmentation and other morphological and biochemical changes in apoptosis [30]. Annexin V binds to phosphatidylserine (PS) in the cell membrane. In normal, live cells, PS is found in the intracellular side of the membrane. During apoptosis, PS is moved to the extracellular side of the membrane which can be marked using annexin V [31]. Propidium iodide is the dye attaching to the DNA of a dead cell, showing the loss of integrity of the cellular membrane. Loss of membrane integrity occurs both, in apoptosis (late phase of this process) and necrosis. 

Four populations of cells present in all tested samples were analyzed by cytometry. Population 1—viable cells which tested negative for annexin V and PI (Figure 3, line A; lower left quadrants on dot plots in Figure 4, column 3, line A and C) and also negative for Caspase-3 and PI (Figure 3, line E; lower left quadrants at dot plots in Figure 4 and Figure 5, column 3, line B and D). 

Population 2—early-apoptotic cells, annexin V positive and PI negative (Figure 3, line B; lower right quadrant, cytogram Figure 4 and Figure 5, column 3, line A and C) and Caspase-3 positive and PI negative (Figure 3, line F; lower right quadrant at dot plot Figure 4, column 3, line B and D). 

Population 3—late-apoptotic cells testing positive for both annexin V and PI (Figure 3, line C; upper right quadrant, cytogram Figure 4, columns 3, line A and C) and also cells positive for Caspase-3 and PI (seen in Figure 3, line G; upper right quadrant at dot plot Figure 4 and Figure 5, column 3, line B and D). 

Population 4—necrotic cells testing positive only for PI (seen in Figure 3, lines D and H; upper left quadrant at dot plot Figure 4 and Figure 5, columns 3, lines A, B, C, D).

Caspase-3 positive cell populations were observed in all cell lines exposed to stilbene derivatives. The lowest percentage of caspase-3 positive cells was observed in CCRF-CEM cell line exposed to rhaponticin (10.07%), HL60 line exposed to deoxyrhaponticin and mitoxantrone (60.82%), HL60/MX1 exposed to rhaponticin (33.70%), HL60/MX2 exposed to rhaponticin (50.3%), and CEM/C1 exposed to rhaponticin and mitoxantrone (85.5%). The highest percentage of caspase-3 positive populations was observed in the CEM/C1 cell line exposed to t-resveratrol, piceatannol, and pterostilbene. A substantial percentage of caspase-3 positive populations was also observed in HL60/MX1 line exposed to t-resveratrol and pterostilbene, HL60 line exposed to *t*-resveratrol and piceatannol, HL60/MX2 line exposed to t-resveratrol and pterostilbene. CCRF-CEM cell line showed the highest resistance (the lowest sensitivity) against tested stilbene derivatives (Figure 6). We have found that the IC_50_ doses for CCRF-CEM cells is low compared to the other cell lines tested, while the number of caspase-3 positive cells is relatively low. In our opinion, this may be due to the high sensitivity of CCRF-CEM cells to the cytotoxic effect of the stilbene derivatives tested, but the induction of the apoptosis process occurs with the participation of another mechanism. An additional explanation for this phenomenon may be a low dose of IC_50_, resulting in a low concentration of stilbene derivatives in the cells, not leading to the induction of apoptosis at the same rate as in the other cell lines.

The comparison of caspase-3 positive cells, detected in two parallel experiments (stilbene vs. stilbene + mitoxantrone) showed that the percentage of caspase-3 positive cells vary. Comparing cells exposed to the stilbene derivative with those exposed to the stilbene derivative with mitoxantrone, there was a significant reduction in the number of apoptotic cells in HL60 cells exposed to deoxyrhaponticin and in CCRF-CEM lines exposed to resveratrol. This phenomenon can be explained by mitoxantrone-activated MDR in tumor cells. In the case of CCRF-CEM cells exposed to rapontycin, a significant increase in the number of apoptotic cells was observed after the addition of mitoxantrone. The remaining samples showed no significant differences (Figure 6).

At the same time, the percentage of cells showing positive detection of annexin V were increased or without significant changes. A significant increase in the number of apoptotic cells in samples exposed to stilbene derivatives and mitoxantrone in comparison to cells exposed only to the stilbene derivative was observed in the case of: rhaponticin in HL60, HL60/MX1, and CCRF-CEM cells; piceatannol and pterostilbene in HL60 cells; resveratrol and deoxyrhaponticin in CCRF-CEM cells. The remaining samples showed no significant changes (Figure 7).

To our knowledge, this is a first published report which presents induction of apoptosis after exposure to rhaponticin (in the presence and absence of mitoxantrone) in HL60, HL60/MX1, HL60/MX2, CCRF-CEM cell lines, deoxyrhaponticin (in the presence and absence of mitoxantrone) in CCRF-CEM and HL60/MX1 cell lines, piceatannol (in the presence and absence of mitoxantrone) in HL60/MX2 cell line, pterostilbene (in the presence and absence of mitoxantrone) in HL60/MX2 which may show a tendency of inhibiting MDR. 

The highest percentage of caspase-3 negative/propidium iodide positive necrotic cells was observed in HL60/MX1, CEM/C1 cell lines exposed to rhaponticin; in HL60 cell line exposed to rhaponticin with mitoxantrone; in HL60, HL60/MX2, CCRF-CEM exposed to piceatannol and in HL60/MX2 cell line exposed to piceatannol with mitoxantrone (Figure 6).

The highest percentage of annexin V negative/propidium iodide positive necrotic cells was observed in HL60/MX1 cell lines exposed to rhaponticin with mitoxantrone, in HL60 cell line exposed to rhaponticin with and without mitoxantrone, in HL60/MX2 exposed to deoxyrhaponticin with and without mitoxantrone (Figure 7).

## 3. Discussion

Plant derivatives (paclitaxel, vinblastine, vinorelbine, vincristine, isothiocyanates, and podophyllotoxin) have been used to treat cancerous diseases for a long time. Naturally occurring stilbenes have attracted the attention of researchers due to extensive and variable biological activity of this group of compounds. Many synthetic derivatives have been developed as well. Antitumor activity of stilbene derivatives has been shown in vitro in many cell lines. The induction of apoptosis in cancer cells by stilbene derivatives is well-documented [17,26,32,33,34,35,36,37,38,39,40,41,42,43]. Stilbene derivatives have been shown to have antitumor activity due to several mechanisms. They are best known for resveratrol and include: ERK1/2 activation, depolarization of the mitchondrial membrane, caspase-3 activation, cell cycle inhibition at the G2/S stage, and inhibition of protein kinase C activity [40,44,45].

In our research, we evaluated stilbene derivatives as a potential antitumor agents and multi-drug resistance modulators. Our results confirmed that RES, PIC, PTER, RHAP, and D-RHAP exhibit antitumor activity, manifested by the induction of apoptosis. It has been already suggested that high concentration of polyphenols (e.g., stilbenes) can induce effective apoptosis [36]. Our studies pointed to specific concentrations of stilbene derivatives which may have an impact on the inhibition of multidrug resistance phenomenon.

IC_50_ doses differ between the cells of the tested lines. CCRF-CEM cells are more sensitive to the effects of the stilbene derivatives tested than the CEM/C1 cells, which are MDR derivatives of the mentioned CCRF-CEM line. This phenomenon was not observed in the case of HL60 cells and its HL60/MX1 and HL60/MX2 derivatives with the MDR phenotype. The cause of these differences can be explained by another type of leukemia from which the mentioned lines originate.

RES has been reported to possess antioxidant activity. What is more remarkable, it also inhibits tumors cells cycle phases by interacting with several cellular targets. Some authors reported RES inducing apoptosis in human T-cell acute lymphoblastic leukemia MOLT-4 cells [37], human monocytic leukemia cell line THP-1 [38], and isolated from patients human B-cells chronic leukaemia cell lines [39]. Opydo-Chanek et al. found that RES and ABT-737 lead to massive apoptosis in MOLT-4 cells [41]. Dörrie et al. found RES-induced apoptosis by depolarizing mitochondrial membranes and activating caspase-3 in human pro-B acute lymphoblastic leukemia cell lines SD, RS4-11, MV4-11, human pre-b leukemia cell lines REH, Nalm-6, and human T-cell leukemia cell lines Jurkat and CEM [40].

It has been shown that dose-dependent PTER induces apoptosis in HL60 cell at concentrations similar to those used in our investigation [32,33,42,43]. Furthermore, our research confirmed that, PTER induces apoptosis in MOLT 4 [43], MV4-11, U937, and THP-1 human leukemia cell lines [32].

PIC was reported as a selective protein-tyrosine kinase inhibitor [46]. PIC leads to cell death via apoptosis of lymphoma cell line BJAB [26] at doses similar to those used in our study. PIC was also reported to induce apoptosis in human monocytic leukemia cells (THP-1 and U937) and human acute myeloblastic cells (HL60) in combination with tumor necrosis factor-related apoptosis-inducing ligand (TRAIL) [34]. 

There are not many studies of apoptotic activity of D-RHAP and RHAP in leukemia cell lines. Our results showed unequivocally that D-RHAP and RHAP lead to massive apoptosis in investigated leukemia cell lines (Figure 6 and Figure 7). We also presented data showing stilbene derivatives inducing apoptosis in cell lines with MDR phenotype, a mechanism which also has not been studied extensively. 

Among the compounds of natural origin with anticancer activity, flavonoids are mentioned [47]. For example, IC_50_ doses determined for quercetin in relation to HL60 lines are in the range of 20–30 μM [48]. The quercetin antitumor mechanisms are inhibition of the cell cycle at G2/M or depolarization of the mitochondrial membrane [49].

Coumarins also show anticancer activity [50]. IC_50_ doses of compounds belonging to the coumarin group are in the following ranges: 15–30 μM for CEM/C1 cells, 10–35 μM for CCRF-CEM cells, 8–60 μM for HL60 cells, 16–70 μM for cell lines HL60/MX2, 20–42 μM for cells of the HL60/MX2 line [51].

Another interesting group of natural compounds with proven antitumor activity are isothiocyanates. This activity consists in inducing the process of apoptosis, inhibiting the cell cycle, or changing the activity of transcription factors like NF-κB or AP-1 [35]. The IC_50_ doses of isothiocyabates determined for the HL60/S line are in the range of 1–4 μM [52], whereas those for the lines with the HL60/ADR and HL60/VCR resistance phenotype are in the 1–10 μM range [53].

Stilbene derivatives are compounds believed to be safe for human consumption [12,54]. Piceatannol and resveratrol show good distribution in tissues, as evidenced by studies in rats, mice, and dogs [54]. Pterostilbene and resveratrol are rapidly absorbed, metabolized (mainly by conjugation with glucuronic acid and sulphate) and excreted in the urine. Pterostilbene has been shown to have stronger pharmacological effects due to its increased stability (due to the difference in structure between pterostilbene and resveratrol–pterostilben has only one hydroxyl group). Attention is paid to the role of intestinal microflora in the level of metabolism of stilbene derivatives in the human body, which is associated with the biotransformation of certain amounts of stilbene derivatives in the lumen of the human intestine by the bacteria present there. Stilbene derivatives are compounds that can potentially be used in treatment, but many studies still need to be carried out in order to better investigate the pharmacokinetics of these compounds and to determine the best route and form of administration [55].

Taken together, our date confirms that t-resveratrol; rhaponticin, deoxyrhaponticin, piceatannol, and pterostilbene have cytotoxic activity and induce apoptosis in HL60, HL60/MX1, HL60/MX2, CCRF-CEM, and CEM/C1 cells lines with and without resistant phenotype. Our study also brings new elements to exploration of stilbene derivatives effect on cancer cells and paves the road to more extensive investigation of this promising direction in alternative cancer treatment options. Among the possibilities of improving the effectiveness of stilbene derivatives therapy, one should mention the possibility of their modification through the creation of new derivatives using stilbene backbone for synthesis [56,57,58]. In addition, new techniques for the administration of active compounds, for example liposomes created using nanotechnology methods, are promising [59].

## 4. Materials and Methods 

### 4.1. Stilbene Derivatives

*t*-Resveratrol (RES), deoxyrhaponticin (D-RHAP), rhaponticin (RHAP), pterostilbene (PTER), and piceatannol (PIC) (Sigma Company, St. Louis, MO, USA) were dissolved in 80% ethanol (Sigma) and their 10 mM concentrations stored at 4 °C.

### 4.2. Cell Lines and Cell Culture

Human acute promyelocytic leukemia cell lines: HL60, HL60/MX1, HL60/MX2, and acute lymphoblastic leukemia cells: CEM/C1 and CCRF-CEM were used in this study. Cell lines were obtained from American Type Culture Collection (ATCC) 10801, University Boulevard, Manassas, VA 20110, USA. 

HL-60 is a promyelocytic cell line derived by S.J. Collins. The peripheral blood leukocytes were obtained by leukopheresis from a 36-year-old Caucasian female with acute promyelocytic leukemia [60]. HL-60/MX1, a mitoxantrone resistant derivative of the HL-60 cell line was obtained from the peripheral blood leukocytes acquired by leukopheresis also from a patient with acute promyelocytic leukemia. HL-60/MX2 is a mitoxantrone resistant derivative of the HL-60 cell line too. HL-60/MX2 cells display atypical multidrug resistance (MDR) with the absence of P-glycoprotein overexpression and altered topoisomerase II catalytic activity including reduced levels of topoisomerase II alpha and beta proteins. CCRF-CEM was derived from human lymphoblasts of the peripheral blood from a child with acute leukemia. CEM/C1 is a camptothecin (CPT) resistant derivative of the human T cell leukemia cell line CCRF-CEM. The cell line was selected and subcloned in 1991 for resistance to CPT. The cells were maintained in RPMI 1640 medium (PAA Laboratories, Linz Austria), supplemented with 10% fetal bovine serum (FBS) (PAA Laboratories) for HL60/MX1, HL60/MX2, CEM/C1, CCRF-CEM cell lines, and 20% FBS for HL60 cells, penicillin-streptomycin (100U/mL PAA Laboratories) and amphotericin (PAA Laboratories) at 37 °C in a humidified atmosphere at 5% CO_2_.

### 4.3. Analysis of Cell Viability

Cells were seeded on 12-well plates (Sarstedt, Wr. Neudorf, Austria) at an initial density of 1 × 10^6^ cells/mL. After 24 h, the cell suspension was stimulated with stilbenes derivatives at concentrations ranging from 10 µM to 1000 µM. After 48 h, 1 mL of cell suspension was centrifuged at 1000 rpm for 5 min, and the supernatant was discarded. The cells were then resuspended in 50 µL phosphate buffered saline (PBS) (PAA Laboratories). From each tube a 10 µL-cell suspension was taken. To each tube 10 µL of Trypan blue reagent (Bio-Rad, Hercules, CA, USA) was added and the sample was incubated for 5 min. Cell viability was measured by Trypan blue assay. Exclusion assay was performed on TC20 Automated Cell Counter (Bio-Rad). Each experiment was repeated three times. 

### 4.4. Leukemic Cells Preparation

The cell samples at 1 × 10^6^ cells/ml density were seeded on 12-well plates. After 24 h, cell lines were treated with stilbene derivatives at the dose of IC_50_ (50% inhibitory concentration). All samples were treated with both stilbene derivatives and mitoxantrone at 20 µM concentration. After 24 h cytometric analysis was performed.

### 4.5. Annexin V-FITC/PI Assay

Cell samples were collected from each well, centrifuged at 1000 rpm for 5 min, and the supernatant was discarded. The cells were resuspended in 50 µL PBS and 50 µL of buffer containing 1 µL annexin V (AnxV), 1 µL propidium iodide (PI) was added and incubated for 15 min at room temperature. Annexin-V-Fluos Staining Kit (Roche Diagnostics, Vienna, Austria) containing annexin V-FITC and propidium iodide (PI) was purchased from Roche. The samples underwent flow cytometric analysis by FlowSight Imaging Flow Cytometer (Amnis, USA) using 488 nm laser. Each experiment was repeated three times.

### 4.6. PhiPhiLux-G_1_D_2_/PI Assay

The cells from each well were collected, centrifuged at 1000 rpm for 5 min, and the supernatant discarded. Then they were resuspended in 750 µL of PBS, centrifuged at 1000 rpm for 5 min, and the supernatant discarded. The cells were resuspended in 20 µL PhiPhiLux G_1_D_2_ containing Caspase3-FITC antibody (OncoImmunin, Gaithersburg, MD, USA) and 1 µL of PI was added. Thus prepared samples were incubated for 60 min at 37 °C. To each tube 1000 µL of binding buffer was added, the samples were centrifuged at 1000 rpm for 5 min, and the supernatant discarded. The samples were analyzed by flow cytometry using FlowSight Imaging Flow Cytometer (Amnis, USA) with 488 nm laser. Each experiment was repeated three times.

### 4.7. FlowSight Cytometric Analysis

The experiments were performed using the FlowSight Imaging Flow Cytometer (Amnis, Seattle, WA, USA) with visualization in real time. The instrument operates as a conventional cytometer but also provides images of every cell tested acting like a fluorescent and inverted microscope. FlowSight is equipped with 488 nm, 658 nm, and 785 nm lasers. In our experiments 488 nm laser was used. At minimum 10,000 cells were measured. Apoptosis analysis was done using Inspire for the FlowSight v.100.2.256.0 flow cytometry analysis software (Amnis). Analysis of apoptosis was performed using Ideas Application v6.0 (Amnis). As a first step, isolated cells were gated on the histogram displaying Channel 1 (BF) Gradient RMS. Next, single cells were gated on dot plot Channel 1 (BF) Area (X) and compared to Channel 1 (BF) Intensity (Y) to exclude aggregated cells. Finally, apoptosis was evaluated on dot plot Channel 2 (X) (green fluorescence, emission range 480–560 nm) referred to Channel 4 (Y) (orange fluorescence, emission range 595–660 nm). Cells without any staining were used as a negative control and there was isotype control applied. Compensation was performed using single stained samples. In our research, FMO (fluorescence minus one) samples were used to determine the gating.

### 4.8. Statistical Analysis

Statistica software v.12.5 (StatSoft, Krakow, Poland) and GraphPad (v.7, GraphPad Software, San Diego, CA, USA). Statistical analysis was accomplished using the one-way Anova with Tukey post-hoc test. Results were considered statistically significant when *p* < 0.05. Data were expressed as mean with standard deviation (SD).

## Figures and Tables

**Figure 1 molecules-24-01896-f001:**
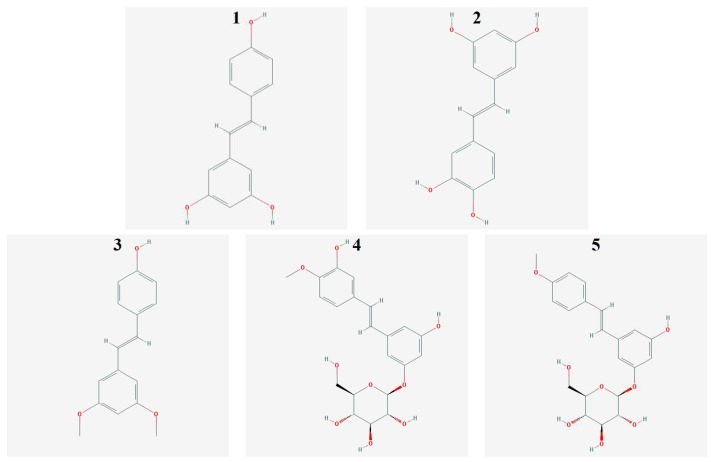
Structural formulas of the stilbene derivatives tested (**1**-*t*-Resveratrol, **2**-Piceattanol, **3**-Pterostilbene, **4**-Rhapontycin, **5**-Deoxyrhaponticin).

**Figure 2 molecules-24-01896-f002:**
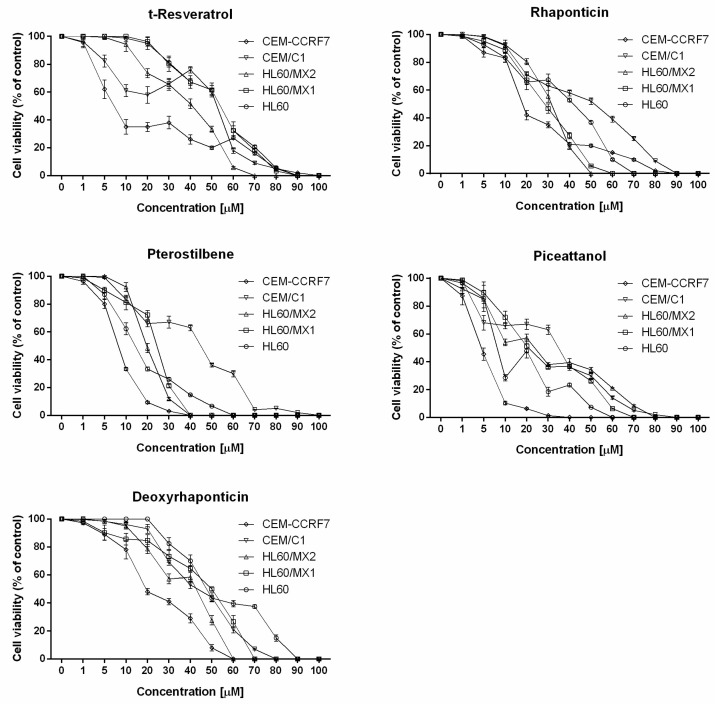
Cell viability analysis of tested stilbene derivatives from conventional trypan blue assay. Each value is shown as mean with standard deviation (SD).

**Figure 3 molecules-24-01896-f003:**
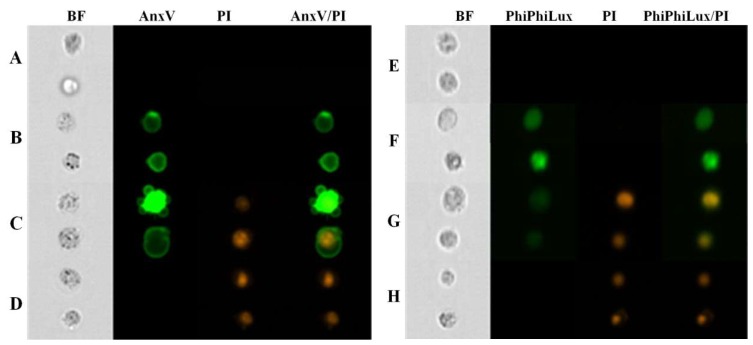
Results of FlowSight analysis of HL60 cells treated with t-resveratrol in IC_50_ (Columns: BF—BrightField, AnxV—Annexin V-FITC, PI—Propidium Iodide; AnxV/PI—Coexpression of Annexin V-FITC and Propidium Iodide; PhiPhiLux—Caspase3-FITC; PhiPhiLux/PI—coexpression of Caspase3-FITC and propidium iodide; Rows: A—viable cells (AnxV-/PI-); B—Early-apoptic cells (AnxV+/PI-); C—Late-apoptic cells (AnxV+/PI+); D—Necrotic cells (AnxV-/PI+); E—viable cells (Casp-/PI-); F— apoptic cells (Casp+/PI-, Casp+/PI+); G—necrotic cells (Casp-/PI+)).

**Figure 4 molecules-24-01896-f004:**
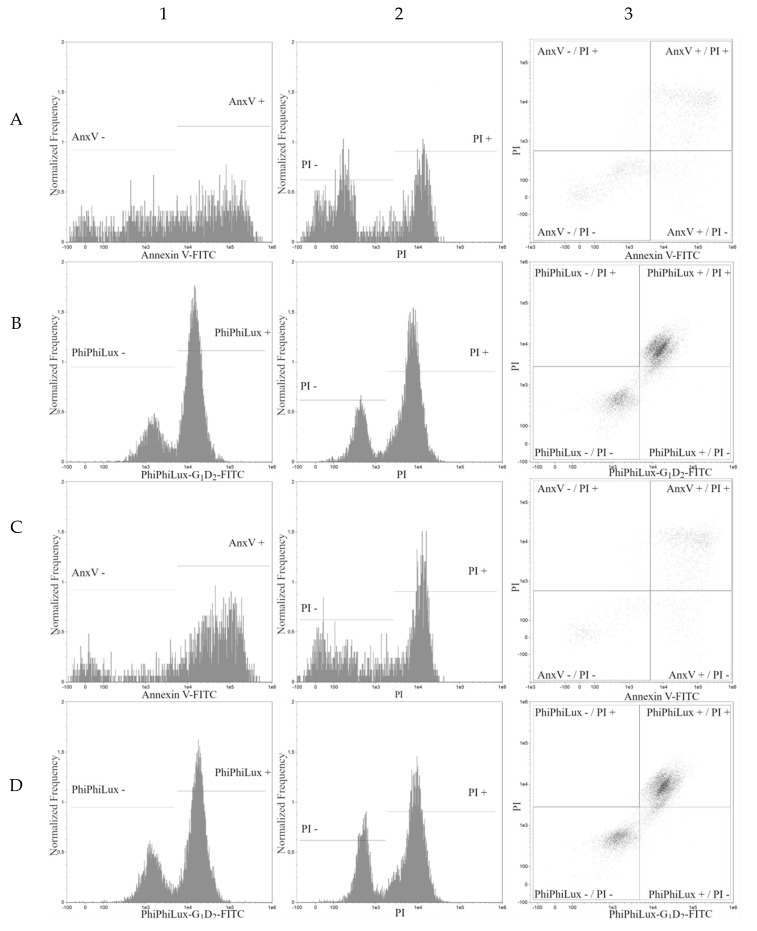
Apoptosis analysis of HL60 cell line. Graphs taken from FlowSight cytometer. All axes are shown in a logarithmic scale. (Columns: 1—Annexin V-FITC and PhiPhiLux-G1D2-FITC histogram; 2—propidium iodide histogram; 3—Annexin V-FITC/propidium iodide dot plot and PhiPhiLux-G1D2-FITC/propidium iodide dot plot; Rows: A, B—IC_50_ of pterostilbene; C, D—IC_50_ of pterostilbene with mitoxantrone. Rows A, C—Annexin V-FITC/propidium iodide assay; B, D—PhiPhiLux-G1D2-FITC/propidium iodide assay).

**Figure 5 molecules-24-01896-f005:**
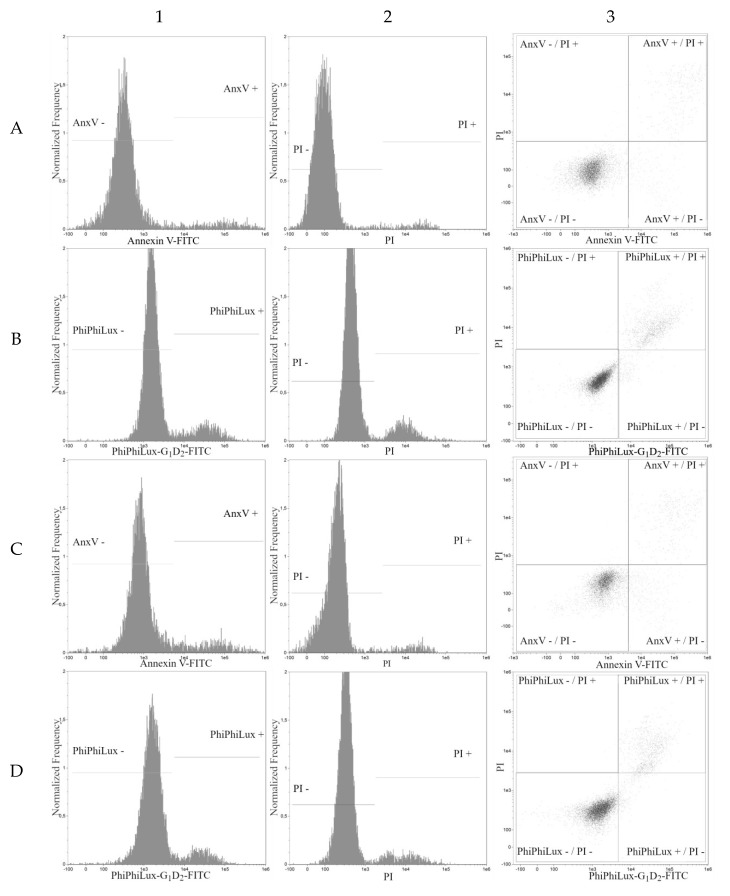
Apoptosis analysis of HL60 cell line. Graphs taken from FlowSight cytometer. All axes are shown in a logarithmic scale. (Columns: 1—Annexin V-FITC and PhiPhiLux-G1D2-FITC histogram; 2—propidium iodide histogram; 3—Annexin V-FITC/propidium iodide dot plot and PhiPhiLux-G1D2-FITC/propidium iodide dot plot; Rows: A, B—untreated HL60 cells; C, D—HL60 cells with mitoxantrone. Rows A, C—Annexin V-FITC/propidium iodide assay, B, D—PhiPhiLux-G1D2-FITC/propidium iodide assay).

**Figure 6 molecules-24-01896-f006:**
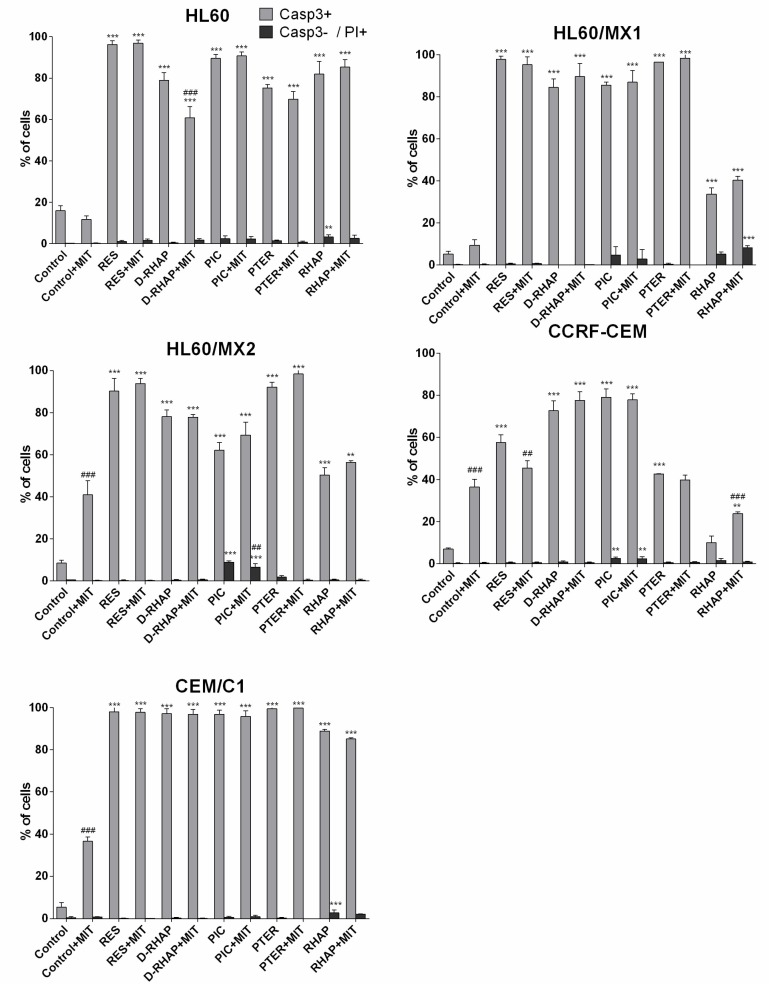
Apoptosis analysis using PhiPhiLux-G_1_D_2_-FITC and propidium iodide on cell lines treated with tested stilbene derivatives. Explanations: Control+MIT—control with mitoxantrone; RES—IC_50_ of resveratrol; RES + MIT—IC_50_ of resveratrol with mitoxantrone; D-RHAP—IC_50_ of deoxyrhaponticin, D-RHAP + MIT—IC_50_ of deoxyrhaponticin with mitoxantrone; PIC—IC_50_ of piceatannol, PIC + MIT - IC_50_ of picetannol with mitoxantrone; PTER—IC_50_ of pterostilbene; D-RHAP + MIT—IC_50_ of pterostilbene with mitoxantrone; RHAP—IC_50_ of rhaponticin; RHAP + MIT—IC_50_ of rhaponticin with mitoxantrone. Each value is shown as Mean ± SD (*n* = 3 per probe; ** *p* < 0.01, *** *p* < 0.001 versus control; ## *p* < 0.01, ### *p* < 0.001 probe with mitoxantrone versus probe without mitoxantrone; one-way ANOVA with Tukey post-hoc test).

**Figure 7 molecules-24-01896-f007:**
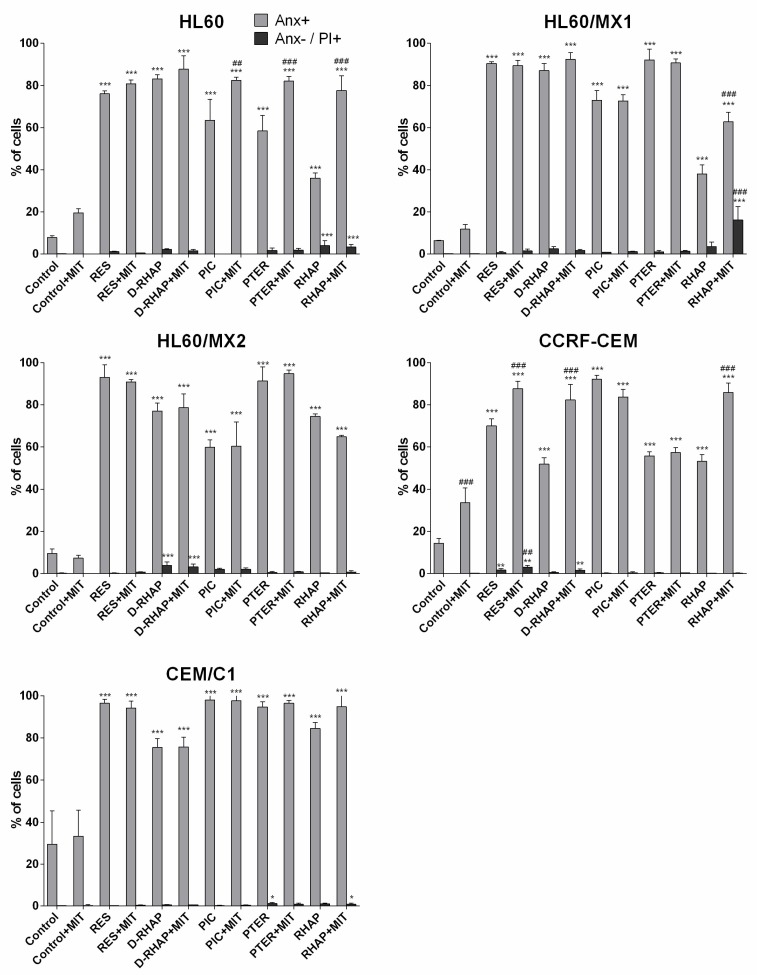
Apoptosis analysis using Annexin V and propidium iodide on cell lines induced with tested stilbene derivatives with absence and presence of mitoxantrone. Explanations: see Figure 6 (*n* = 3 per probe; * *p* < 0.05, ** *p* < 0.01, *** *p* < 0.001 versus control; ## *p* < 0.01, ### *p* < 0.001 probe with mitoxantrone versus probe without mitoxantrone; one-way ANOVA with Tukey post-hoc test).

**Table 1 molecules-24-01896-t001:** IC_50_ determined on tested cell lines. Each dose is shown in micromolar concentration [µM] as Mean ± SD.

Cell Line	*t*-Resveratrol (Mean ± SD)	Deoxyrhaponticin (Mean ± SD)	Pterostilbene (Mean ± SD)	Rhapontin (Mean ± SD)	Piceatannol (Mean ± SD)
HL60	54.09 ± 5.69	47.54 ± 3.90	14.25 ± 1.48	41.75 ± 2.33	8.12 ± 2.17
HL60/MX1	53.95 ± 1.18	50.52 ± 2.46	24.35 ± 2.57	28.38 ± 1.26	20.76 ± 2.15
HL60/MX2	40.88 ± 1.85	42.71 ± 2.60	19.62 ± 1.95	31.51 ± 2.51	23.73 ± 1.63
CEM/C1	52.38 ± 3.28	43.33 ± 3.59	44.82 ± 2.75	51.54 ± 2.36	34.82 ± 3.87
CCRF-CEM	7.22 ± 1.78	19.33 ± 1.51	8.21 ± 0.72	18.05 ± 2.22	4.57 ± 0.12

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
