# Peer review of "Imaging Flow Cytometric Analysis of Stilbene-Dependent Apoptosis in Drug Resistant Human Leukemic Cell Lines"

_molecules, 2019, doi:10.3390/molecules24101896_

Round 1

Reviewer 1 Report

This manuscript analyzed the effects of stilbene derivatives on drug resistant human leukemic cells, and found the investigated stilbene derivatives have various influence on the cancer cell viabilities. This is a very interesting study addressing the effects of these stilbenes on the cell viability and apoptosis of the caspase-3 positive cells with mitoxantrone-induced resistance. The experimental procedures were reported completely and the data are clearly presented and support the conclusions.
A couple of minor concerns includes:
1.    The activities of these stilbene derivatives were not measured in the normal cells. Since the high concentrations (>50uM) of the stilbene derivatives were used to in this study, these compounds should be tested on normal cell lines to avoid toxicities.
2.    The discussion part in the manuscript summarized the research results and compared the current results to the previously reported data, but it would be better if the authors can discuss more about the potential mechanisms of these compounds involved in the cell apoptosis.

Author Response

Thank you very much for the review.

Below are the answers:

1.       The activities of these stilbene derivatives were not measured in the normal cells. Since the high concentrations (>50uM) of the stilbene derivatives were used to in this study, these compounds should be tested on normal cell lines to avoid toxicities. 
During the development of the data an error crept in which caused a wrong conversion of doses. This means that correctly calculated doses are 10 times smaller than those found in the original version of the manuscript.

In our opinion, there is no need to analyze the effect of test compounds in designated doses on human normal cells.

2.       The discussion part in the manuscript summarized the research results and compared the current results to the previously reported data, but it would be better if the authors can discuss more about the potential mechanisms of these compounds involved in the cell apoptosis.

We added more information on the activity of stilbene compounds at work.

Reviewer 2 Report

Authors evaluated the apoptotic effect of stilbene derivatives on leukemic cells. Indeed all derivatives tested seems to induce cell apoptosis, based mainly on caspase-3 activation and annexin V positive staining. However results are presented in a confusing manner and some of the are not completely supported by the showed data.

All stilbene derivatives seem to have a low citotoxicity on tested cell lines (most IC50 are above 0.3-0.4 mM) with the exception of CCRF-CEM cell line (table 1 and figure2). Authors do not comment, neither evidence this behavior. In addition, CCRF-CEM cells are much more sensitive to  stilbene derivatives compared to CEM/C1 cells, corresponding to the same lymphoblastic leukemia cell line but resistant to mitoxantrone. This is not true for acute promyelocytic leukemia cell lines ether resistant or not to mitoxantrone (HL60 vs HL60/MX1 and HL60/MX2), which show a similar sensitivity to stilbene derivatives. Do the Authors have an explanation of this. It will be valuable extend this observation to other leukemia cell lines.  

Figure 3 seems unnecessary, while in figure 4 is missing the control cell line (HL60), untreated with stilbene derivate. This will help to justify the position of gates used to distinguish positive from negative cells, in particular for annexin V-FITC staining (In HL60 cell treated with pterostilbene it is hard to distinguish negative from positive cells). A control untreated HL60 cells will make easy make such distinction. Additionally, as I shall state below, the increase in annexin V-FITC positive cells after co-treatment with pterostilbene and mitoxantrone (MIT) seems the result of  apoptosis induced by mitoxantrone itself (in agreement with its topoisomerase II inhibitory activity) rather than the co-treatment (this is evident in figure 6, HL60 cells treated with MIT alone).

Considering figures 5 and 6, to make a distinction between early (Casp3+/PI-) and late (Casp3+/PI+) apoptotic cells can create some confusion. In some case the increase of the early apoptotic population is accompanied by a decrease in the late one and vice versa. So, in my opinion, it would be better to consider the overall apoptotic population and compare it in the different cell lines treated with stilbene derivatives. Additionally, most of the tested condition mainly result in a late apoptotic state (more than 60% of cell population) and early apoptotic cells rarely are above 5%.

Authors state that the percentage of caspase-3 positive cells is decreased or unchanged in samples exposed to MIT and  stilbene derivatives (lines 211-212). I cannot see this. In most of the cases it cannot be observed an appreciable variation in caspase-3 positive cells exposed to MIT and  stilbene derivatives compared to cells exposed only to stilbene derivatives. On the contrary, there are several cases where co-treatment result in an increase of caspase-3 positive cells (RHAP in HL60/MX1, HL60MX2 and CCRF-CEM cell lines, PIC and PTER in HL69/MX2 cells) see figure 5. This can be the result of apoptosis induced by MIT.

In addition, and more important, all consideration made by the Authors about differences in the percentage of apoptotic cells between treatment with stilbene derivatives (in absente or presence of MIT) are not based on a robust statistical analysis (Student’s t-test was performed analyzing all treated samples versus control/untreated cells). A different statistical analysis (i.e. ANOVA), to compare differences between cell treatment with stilbene derivatives in absente or presence of MIT, would be more appropriate.

Authors state that the number of  annexin-V positive cells is decreased in samples exposed to MIT and  stilbene derivatives compared to cells exposed to stilbene derivatives alone, with the exception of RES and D-RHAP (lines 213-216). Again, I cannot clearly see this. In most of the case, co-treatment with MIT and  stilbene derivatives results in an increase of annexin-V positive cells (D-RHAP, PIC, PTER and RHAP in HL60 cells, RES, D-RHAP and RHAP in CCRF-CEM cells, and RHAP in CEM/C1 cells) see figure 6.

All cyto-fluorimetric determination were performed on cells treated with an IC50 concentration of stilbene derivatives. This could result in a low cell viability and thid is supported by the very high percentage of late apoptotic cells observed in most of the treated cells. No mention is made about the number of cell subjected to apoptotic analysis and the number of surviving cells after different treatments compared to the untreated ones. I can suggest to use a stilbene derivatives dosage below IC50%.

Overall all stilbene derivatives can induce apoptosis in all tested cell lines, with a high extent in the majority of the conditions. Additionally, no appreciable difference between treatment with stilbene alone or in the presence of MIT can be observed in most of the cases. For some there is a difference in the level of apoptosis induced with stilbene alone or in the presence of MIT but  these differences, either increase or decrease,  are not supported by a statistical analysis.

To improve the quality of the manuscript I suggest the Authors to extensively revise it, with particular attention to the result section, and provide additional information.

Author Response

Thank you very much for the review.

Below are the answers:

1.     All stilbene derivatives seem to have a low citotoxicity on tested cell lines (most IC50 are above 0.3-0.4 mM) with the exception of CCRF-CEM cell line (table 1 and figure2). Authors do not comment, neither evidence this behavior.

During the development of the data an error crept in which caused a wrong conversion of doses. This means that correctly calculated doses are 10 times smaller than those found in the original version of the manuscript.

2.     In addition, CCRF-CEM cells are much more sensitive to  stilbene derivatives compared to CEM/C1 cells, corresponding to the same lymphoblastic leukemia cell line but resistant to mitoxantrone. This is not true for acute promyelocytic leukemia cell lines ether resistant or not to mitoxantrone (HL60 vs HL60/MX1 and HL60/MX2), which show a similar sensitivity to stilbene derivatives. Do the Authors have an explanation of this. It will be valuable extend this observation to other leukemia cell lines.  

An attempt to explain this phenomenon has been included in the work.

“IC50 doses differ between the cells of the tested lines. Cells of the CCRF-CEM line are more sensitive to the cytotoxic effect of the stilbene derivatives tested than the CEM/C1 cells, which are MDR derivatives of the mentioned CCRF-CEM line. This phenomenon was not observed in the case of HL60 cells and its HL60/MX1 and HL60/MX2 derivatives with the MDR phenotype. The cause of these differences can be explained by another type of leukemia from which the mentioned lines originate.”

3.     Figure 3 seems unnecessary, while in figure 4 is missing the control cell line (HL60), untreated with stilbene derivate. This will help to justify the position of gates used to distinguish positive from negative cells, in particular for annexin V-FITC staining (In HL60 cell treated with pterostilbene it is hard to distinguish negative from positive cells). A control untreated HL60 cells will make easy make such distinction.

In our opinion, the pictures show (prove) that apoptosis occurs in the examined cells. Of course, they can be removed from work. We have added graphs for controls and controls exposed to mitoxantrone to work.

4.     Additionally, as I shall state below, the increase in annexin V-FITC positive cells after co-treatment with pterostilbene and mitoxantrone (MIT) seems the result of  apoptosis induced by mitoxantrone itself (in agreement with its topoisomerase II inhibitory activity) rather than the co-treatment (this is evident in figure 6, HL60 cells treated with MIT alone).

5.     Considering figures 5 and 6, to make a distinction between early (Casp3+/PI-) and late (Casp3+/PI+) apoptotic cells can create some confusion. In some case the increase of the early apoptotic population is accompanied by a decrease in the late one and vice versa. So, in my opinion, it would be better to consider the overall apoptotic population and compare it in the different cell lines treated with stilbene derivatives. Additionally, most of the tested condition mainly result in a late apoptotic state (more than 60% of cell population) and early apoptotic cells rarely are above 5%.

The proposed change has been included. New graphs showing only apoptotic and necrotic cells were placed at work.

6.     Authors state that the percentage of caspase-3 positive cells is decreased or unchanged in samples exposed to MIT and  stilbene derivatives (lines 211-212). I cannot see this.

In most of the cases it cannot be observed an appreciable variation in caspase-3 positive cells exposed to MIT and  stilbene derivatives compared to cells exposed only to stilbene derivatives.

On the contrary, there are several cases where co-treatment result in an increase of caspase-3 positive cells (RHAP in HL60/MX1, HL60MX2 and CCRF-CEM cell lines, PIC and PTER in HL69/MX2 cells) see figure 5. This can be the result of apoptosis induced by MIT.

These conclutions are modified.

7.     Authors state that the number of  annexin-V positive cells is decreased in samples exposed to MIT and  stilbene derivatives compared to cells exposed to stilbene derivatives alone, with the exception of RES and D-RHAP (lines 213-216). Again, I cannot clearly see this. In most of the case, co-treatment with MIT and  stilbene derivatives results in an increase of annexin-V positive cells (D-RHAP, PIC, PTER and RHAP in HL60 cells, RES, D-RHAP and RHAP in CCRF-CEM cells, and RHAP in CEM/C1 cells) see figure 6.

Of course, this is a good point and these conclusions have been changed.

8.     All cyto-fluorimetric determination were performed on cells treated with an IC50 concentration of stilbene derivatives. This could result in a low cell viability and thid is supported by the very high percentage of late apoptotic cells observed in most of the treated cells. No mention is made about the number of cell subjected to apoptotic analysis and the number of surviving cells after different treatments compared to the untreated ones. I can suggest to use a stilbene derivatives dosage below IC50%.

In our studies, we also determined IC10 doses, however, we decided that at work we would only present the results of analyzes for IC50 doses. The number of results that we present only for IC50 doses is quite high in our opinion.

9.     Overall all stilbene derivatives can induce apoptosis in all tested cell lines, with a high extent in the majority of the conditions. Additionally, no appreciable difference between treatment with stilbene alone or in the presence of MIT can be observed in most of the cases. For some there is a difference in the level of apoptosis induced with stilbene alone or in the presence of MIT but  these differences, either increase or decrease,  are not supported by a statistical analysis.

Another method of statistical analysis (one-way Anova) was used, which gave the opportunity to assess the significance of differences between samples exposed to the stilbene derivative in comparison to the sample exposed to the stilbene derivative and mitoxantrone.

Reviewer 3 Report

In this manuscript, CZOP et al., studied the effects of a few natural compounds (resveratrol, piceatannol, rhaponticin, deoxyrhaponticin etc.) on the induction of apoptosis in drug resistant Human Leukemic Cell Lines using Flow Cytometry.

The main concern from this reviewer is the data in Table 1 - the IC50 determined on tested cell lines. It seems these Stilbene derivatives are not biologically active in the induction of cell death, most IC50 values are >100uM. 

The authors could compare their bioactivity and effectiveness with known anti-cancer drugs. For example, IC50 of cisplatin in Hl-60 was only 1-2uM (see Su WC et al., Jpn J Clin Oncol 2000;30(12).

Isothiocyanates have IC50 values below 10uM (see Zhang Y et al., Selected isothiocyanates rapidly induce growth inhibition of cancer cells. Mol Cancer Ther. 2003;2(10):1045-52. Jakubikova J et al., Isothiocyanates induce cell cycle arrest, apoptosis and mitochondrial potential depolarization in HL-60 and multidrug-resistant cell lines. Anticancer Res. 2005;25(5):3375-86. Lenzi M et al., 6-(Methylsulfonyl) hexyl isothiocyanate as potential chemopreventive agent: molecular and cellular profile in leukaemia cell lines. Oncotarget. 2017; 8:111697-111714). Quercetin has IC50 values about 10-20uM (Kang TB, Liang NC. Biochem Pharmacol. 1997;54(9):1013-8). Recently, more active isoflavone ME-344 was shown to have an IC50 in the range of 70-260 nM (Jeyaraju DV, Oncotarget. 2016;7(31):49777-49785).

Lines 272-273 “Taken together, our date confirms that t-resveratrol; rhaponticin, deoxyrhaponticin, piceatannol and pterostilbene have anticancer potential…” More experiments and mechanistic studies are needed to confirm their anti-cancer effects.

Author Response

Thank you very much for the review.

Below are the answers:

1.     The main concern from this reviewer is the data in Table 1 - the IC50 determined on tested cell lines. It seems these Stilbene derivatives are not biologically active in the induction of cell death, most IC50 values are >100uM. 

During the development of the data an error crept in which caused a wrong conversion of doses. This means that correctly calculated doses are 10 times smaller than those found in the original version of the manuscript.

2.     Lines 272-273 “Taken together, our date confirms that t-resveratrol; rhaponticin, deoxyrhaponticin, piceatannol and pterostilbene have anticancer potential…” More experiments and mechanistic studies are needed to confirm their anti-cancer effects.

Of course, the statement that the compounds tested have anticancer potential is far-reaching. This statement has been changed to the information that the stilbene derivatives studied have cytotoxic activity and induce apoptosis of the cells of the tested lines.

Round 2

Reviewer 2 Report

Authors revised the manuscript but still some issues remain.

There is an inconsistency in CCRF-CEM line between the reported high sensitivity to all stilbene derivatives  (low IC50 values, see table 1) and the apparent resistance based  mainly on the low percentage of casp3 positive cells observed (see figures 6 and 7) justified as a low sensitivity to these agents (line 220). How is it possible to explain this behavior. Stilbene derivatives cytotoxicity reported on CCRF-CEM line may be the result of a different cellular mechanism or the apoptotic effect may require longer time (considering the lower concentration of substance utilized on this cell line). Authors should at least comment this.

Overall all stilbene derivatives can induce apoptosis in all tested cell lines (compared to both untreated and MIT only treated cells), with a high extent in the majority of the conditions. Additionally, no appreciable difference between the treatment with stilbene alone or the co-treatment in the presence of MIT can be observed in most of the cases, with the exception of CCRF-CEM cells where the co-treatment results in an increased apoptotic effect compared to stilbene derivative alone. However, sentence in lines 245-249 “..this is a first published report which presents a proportionate increase in the number of casp3 positive cells…” is not clear and seems to be confusing. The “proportionate increase in the number of casp3 positive cells” is attributed to selected combination of stilbene derivates and cell lines which, furthermore, are not  clearly evident (Figure 6 and 7) and not supported by a statistical significance. In my opinion this sentence should be modified or made cleaner. If the Author refer to a “first report” it would be better to specify  D-RHAP and RHAP for which there are not may studies on their apoptotic activity, in particular the case of MDR cell lines (as stated in discussion, see lines 306-309).

A minor point (line 155 concentration need to be corrected, 100-220 uM -> 10-20 uM)

Author Response

1. There is an inconsistency in CCRF-CEM line between the reported high sensitivity to all stilbene derivatives (low IC50 values, see table 1) and the apparent resistance based mainly on the low percentage of casp3 positive cells observed (see figures 6 and 7) justified as a low sensitivity to these agents (line 220). How is it possible to explain this behavior. Stilbene derivatives cytotoxicity reported on CCRF-CEM line may be the result of a different cellular mechanism or the apoptotic effect may require longer time (considering the lower concentration of substance utilized on this cell line). Authors should at least comment this.

Thank you very much for drawing attention to this interesting relationship, which, of course, is worth describing. Below is our explanation of this phenomenon.

“We have found that the IC50 doses for CCRF-CEM cells is low compared to the other cell lines tested, while the number of caspase-3 positive cells is relatively low. In our opinion, this may be due to the high sensitivity of CCRF-CEM cells to the cytotoxic effect of the stilbene derivatives tested, but the induction of the apoptosis process occurs with the participation of another mechanism. An additional explanation for this phenomenon may be a low dose of IC50, resulting in a low concentration of stilbene derivatives in the cells, not leading to the induction of apoptosis at the same rate as in the other cell lines.”

2.Overall all stilbene derivatives can induce apoptosis in all tested cell lines (compared to both untreated and MIT only treated cells), with a high extent in the majority of the conditions. Additionally, no appreciable difference between the treatment with stilbene alone or the co-treatment in the presence of MIT can be observed in most of the cases, with the exception of
CCRF-CEM cells where the co-treatment results in an increased apoptotic effect compared to stilbene derivative alone. However, sentence in lines 245-249 “..this is a first published report which presents a proportionate increase in the number of casp3 positive cells…” is not clear and seems to be confusing. The “proportionate increase in the number of casp3 positive cells” is attributed to selected combination of stilbene derivates and cell lines which, furthermore, are not clearly evident (Figure 6 and 7) and not supported by a statistical significance. In my opinion this sentence should be modified or made cleaner. If the Author refer to a “first report” it would be better to specify D-RHAP and RHAP for which there are not may studies on their apoptotic activity, in particular the case of MDR cell lines (as stated in discussion, see lines 306-309).

Thank you for paying attention to this issue. This result has been modified:
“To our knowledge, this is a first published report which presents induction of apoptosis after exposure to rhaponticin (in the presence and absence of mitoxantrone) in HL60, HL60/MX1, HL60/MX2, CCRF-CEM cell lines, deoxyrhaponticin (in the presence and absence of mitoxantrone) in CCRF-CEM and HL60/MX1 cell lines, piceatannol (in the presence and absence of mitoxantrone) in HL60/MX2 cell line, pterostilbene (in the presence and absence of mitoxantrone) in HL60/MX2 which may show a tendency of inhibiting MDR.”

3.A minor point (line 155 concentration need to be corrected, 100-220 uM -> 10-20 uM)

This is an oversight created during the correction of the manuscript. This has obviously been improved.

Reviewer 3 Report

In this revised manuscript, the authors have demonstrated that natural compounds (stilbene derivatives), possess cytotoxic activities and induce apoptosis in human leukemic cells lines with and without resistant phenotype. These natural compounds are promising in alternative medicine and/or cancer management. 

Author Response

Dear Reviewer,

Thank you very much for the review.
